# The impact of environmental regulation on green total factor productivity: An empirical analysis

Qin He[1], Yaowu Han[2]*, Lei Wang[1]

1 School of Public Administration, Hunan Normal University, Changsha, Hunan Province, China, 2 School of Mechanical Engineering, Hunan Institute of Engineering, Xiangtan, Hunan Province, China

* hanyaowuhie@163.com

## Abstract

The transformation of China's economy from extensive growth to high-quality development is essentially an increase in green total factor productivity (GTFP). China currently has a range of environmental regulation tools, and the question of whether environmental regulation can promote improvement in China's GTFP requires theoretical and empirical analysis. This article first divides environmental regulation into three types: administrative, market-based and information-based. It then builds an empirical model of the effect of environmental regulation on GTFP. Slacks based measure-data envelope analysis (SBM-DEA) and the Malmquist index are used to measure the GTFP of 30 provinces in China from 2005 to 2018, and a measurement model of the impact of environmental regulation on GTFP is established. The results show that: (1) there are significant differences in GTFP in eastern, central and western China; (2) there is a non-linear relationship between environmental regulations and GTFP.

## 1. Introduction

Since the economic program known as "reform and opening-up," China's traditional extensive economic methods have brought rapid economic growth through aggregate advantages. However, this growth has inevitably caused large amounts of natural resource consumption and environmental pollution, affecting and restricting the future opportunities for sustainable economic and social development. BP's World Energy Statistical Yearbook [1] shows that in 2017 China accounted for 23.2% of global energy consumption and 33.6% of global energy consumption growth, making it the world's largest energy consumer, ranked first in global energy growth for 17 consecutive years. At the same time, in the latest Global Environmental Performance Index (EPI) ranking released in 2018, China ranked 120th out of 180 participating countries and regions, and was fourth lowest for air quality. The frequent occurrence of haze has seriously affected people's lives and health and has become a major concern of the Chinese people.

Since the 18th National Congress of the Communist Party of China, the Chinese government has realized the unsustainability of extensive economic growth, and has been working on

**Data Availability Statement:** All date files are available from the Statistical Yearbook of China (2005-2018) and Environment Statistical Yearbook of China(2005-2018). They are third party data that anyone can access this data in the same manner as

the authors through http://www.stats.gov.cn/tjsj./ndsj/.

**Funding:** This work was supported by the Philosophy and Social Science Foundation of Hunan Province, China (Grant No. 20YBA160). Qin He received this award. The funders had no role in study design, data collection and analysis, decision to publish, or preparation of the manuscript.

**Competing interests:** The authors declare that they have no conflict of interest.

environmental pollution control and achieving high-quality economic growth. The report of the 19th National Congress in 2017 made a major new judgment on the Chinese economy, namely, that it has shifted from high-speed growth to high-quality development. Transformation has become the consensus of China's economic reform. The report emphasized the need for unswerving implementation of the five development concepts of "innovation, coordination, greenness, openness, and sharing" and for the establishment of a modern economic system with a green, low-carbon cycle. Economic green transformation requires resource conservation and environmental protection. It involves the process of economic development gradually moving toward "increased labor productivity, reduced pollution emissions, reduced resource energy consumption, and enhanced sustainable development capabilities." Essentially, this means continuous improvement in green total factor productivity (GTFP). Therefore, under the new normal, GTFP is a necessary condition for the implementation of hard constraints on resources and the environment [2]. Given the importance of improving the quality of economic growth and achieving green development, the task of increasing the contribution of GTFP to economic growth has become urgent for China [3].

Shen et al. [4] pointed out that the implementation of appropriate administrative and market-based environmental regulatory policies to promote the improvement of GTFP has become an important element in the current green transition of China's economy, and it is also a problem that China urgently needs to solve. In recent years, China has introduced a series of different types of environmental regulation tools. Peng [5] suggested that the formulation and enforcement of the current environmental regulations restrain economic growth, and that this has led to a continuous increase in the social welfare costs of those regulations. In this context, it is necessary to focus on a number of related issues: whether and how environmental regulation can promote China's GTFP to achieve a win-win situation for environmental pollution control and high-quality economic growth; the different effects of different regulatory tools on GTFP; and changes in China's GTFP following the increase in environmental regulation. This article attempts to address these issues, providing a scientific basis for assessing the effectiveness of China's existing environmental regulatory policies and selecting policy tools that reflect regional economic development.

Existing research mostly adopts a comprehensive variable to reflect the strength of environmental regulation. This article considers the advantages and application of different types of environmental regulation tools, categorizing environmental regulation into three types: administrative, market-based and information-based. The possible non-linear relationship between different types of environmental regulations and GTFP is tested theoretically and empirically. Taking into account the heterogeneity existing in different regions, this provides a scientific basis for choosing environmental regulatory policy tools that are compatible with regional economic development [6–8].

Drawing on the DEA-Malmquist total factor productivity model and examining the impact of environmental regulations on GTFP, this article further clarifies that the effect of environmental regulation on GTFP depends on regional differences [9]. It therefore provides a reference for developing countries seeking to carry out environmental regulation and improve TFP.

## 2. Literature review

Tietenberg [10] classified environmental regulatory policy tools into three categories according to development time: order-control, market-based and voluntary. The first is a command-and-control tool, which is mainly used by governments for environment-related laws, regulations and standards. This tool is widely used and has obvious effects. It was first used by the

United States, Japan and European countries and is relatively mature. Among the most widely used environmental regulatory tools of this type are measures to establish environmental pollution emission standards and to limit the concentration and volume of pollutants emitted by enterprises [11]. The second is a tool that relies mainly on market regulation mechanisms to encourage companies to participate in environmental pollution control. Common market-type regulatory tools include sewage charges, pollution permit transactions, environmental technology innovation and financial subsidies [12,13]. Compared to command-and-control tools, market tools can sometimes lead to unexpected results and lower costs [14,15]. The third tool is a voluntary tool, which Khanna et al. [16] called the "third wave" of environmental regulation. Its role is based on information disclosure and public participation, including environmental letters and visits, resource agreements and eco-labels. Becker et al. [17] and Boyer [18] observed that a large part of the enforcement work of the US Environmental Protection Agency relies on private lawsuits. Arimura et al. [19] found that the use of an environmental performance white paper is conducive to improving the efficiency of use of natural resources and thus to improving environmental quality.

Environmental regulations do not have a uniform fixed model, and standards of environmental regulation vary. This creates significant measurement difficulties, but four main types of indicators can be identified. (1) Cost indicators are the most common, and include sewage charges and environmental taxes. [20–24] (2) Input indicators include environmental protection-related fiscal expenditures and investment in environmental pollution treatment. [25–27] (3) Performance indicators, such as pollutant emissions and carbon emissions per unit of industrial output value, reflect the effects of environmental regulations in terms of governance environmental performance. [28–31] (4) A comprehensive index was first used by Walter and Ugelow [32] as a measure of environmental regulation. Since then, this type of index has been widely used by scholars for its comprehensive coverage of a wide range of characteristics.

DEA has often been used to measure China's environmental efficiency and GTFP [33]. Song and Wang [34] examined environmental efficiency at the national level, dividing the factors that affect environmental efficiency into two categories (technical factors and environmental regulations) and using these to quantify environmental regulations. Li, H, et al. [35] used the Super-SBM model of undesired output to measure China's environmental efficiency from 1991 to 2010, including a Tobit regression model to explore the relevant factors. Song et al. [36] measured the efficiency of environmental regulation following China's accession to the WTO, exploring the factors affecting the environmental efficiency of different provinces, and using various prediction models to predict environmental efficiency from 2011 to 2012. Long et al. [37] analyzed the impact of China's accession to the WTO on environmental policies and found that, although China has adopted stricter regulations to meet higher standards, accession alone does not guarantee better environmental conditions. Yang et al. [38] measured the environmental efficiency of 30 provinces in China from 2000 to 2010 using a super-efficiency DEA model, and other scholars have studied environmental efficiency at the regional level [39,40]. In the current literature, industrial and regional environmental efficiency are collectively referred to as environmental efficiency [41] or as GTFP [42]. In measuring GTFP, the effects of poor output are fully considered. Most scholars [4,43] use labor and fixed asset investment as input indicators and gross domestic product (GDP) as indicators. This article uses the SBM-Undesirable model to measure GTFP. Because model benefits from non-radial and non-directional characteristics, it is fully capable of measuring the error caused by bad output, thereby minimizing bad output and allowing the use of the Malmquist index [44].

Building on this existing literature, the present paper uses the data envelopment method of analyzing undesired output to incorporate energy consumption and environmental pollution emissions into the total factor productivity measurement system, thereby measuring the value

of GTFP and exploring the internal structure and motivation of GTFP growth. Environmental regulations are subdivided into three types: administrative, market-based and information-based. The relationship between environmental regulations and GTFP and its decomposition items is empirically tested. In addition, given that levels of resource endowment, industrial structure and development stages vary across the regions of China and that the regional impacts of environmental regulations on GTFP are therefore different, three regions are considered: the eastern, the central, and the western regions. Comparative analysis is used to examine the growth differences in GTFP between regions and to formulate appropriate environmental regulatory policy tools based on regional levels of economic development.

## 3. Model construction and data description

### 3.1. Econometric model

This article uses data from 30 provinces (cities) in China from 2005 to 2018. The data meet the requirements of the panel data model. The basic definition of the panel regression model is:

$$GTFP_{it} = \alpha + \beta_0 GTFP_{i,t-1} + \beta_1 ER_{i,t} + \beta_j Control_{it} + \varepsilon_{it} \tag{1}$$

where $i$ and $t$ are the individual and the year, $\alpha$ is a constant term, $\beta$ is a parameter to be estimated, $\varepsilon_{it}$ is a random error term, $GTFP$ is the green total factor productivity, $GTFP_{i,t-1}$ is the lag period of the explained variable, $ER$ is the environmental regulation, and $Control$ denotes the control variables. The environmental regulation $ER$ includes three effects, namely, government-commanded environmental regulation ($Gover$), market-based environmental regulation ($Market$), and information-based environmental regulation ($Inform$). Model (1) can therefore be expressed as follows:

$$GTFP_{it} = \alpha + \beta_0 GTFP_{i,t-1} + \beta_1 Gover_{i,t} + \beta_j Control_{it} + \varepsilon_{it} \tag{2}$$

$$GTFP_{it} = \alpha + \beta_0 GTFP_{i,t-1} + \beta_1 Market_{i,t} + \beta_j Control_{it} + \varepsilon_{it} \tag{3}$$

$$GTFP_{it} = \alpha + \beta_0 GTFP_{i,t-1} + \beta_1 Inform_{i,t} + \beta_j Control_{it} + \varepsilon_{it} \tag{4}$$

The impact of environmental regulation on GTFP may not be a simple linear relationship. In order to test the hypothesis of non-linear effects on GTFP of the three types of environmental regulation (administrative, market-based and information-based), environmental regulations are introduced into the measurement model, which is represented as follows:

$$GTFP_{it} = \alpha + \beta_0 GTFP_{i,t-1} + \beta_1 Gover_{i,t} + \beta_2 (Gover_{i,t})^2 + \beta_j Control_{it} + \varepsilon_{it} \tag{5}$$

$$GTFP_{it} = \alpha + \beta_0 GTFP_{i,t-1} + \beta_1 Market_{i,t} + \beta_2 (Market_{i,t})^2 + \beta_j Control_{it} + \varepsilon_{it} \tag{6}$$

$$GTFP_{it} = \alpha + \beta_0 GTFP_{i,t-1} + \beta_1 Inform_{i,t} + \beta_2 (Infrom_{i,t})^2 + \beta_j Control_{it} + \varepsilon_{it} \tag{7}$$

### 3.2. Variable selection and data description

**3.2.1. GTFP.** This article uses the SBM-DEA model with undesired outputs to calculate GTFP. The method assumes that there are $n$ decision making units (DMU) in the production system where each DMU has inputs and outputs. Each DMU has three vectors, including an input vector and two output vectors. Input vectors are set as $X = [x_1, x_1, \cdots x_n] \in R^{m \times n}$,

desirable output vectors as $Y^g = [y_1^g, y_2^g, \cdots y_n^g] \in R^{s_1 \times n}$ and undesirable output vectors as $Y^b = [y_1^b, y_2^b, \cdots y_n^b] \in R^{s_2 \times n}$. If $X > 0$, $Y^g > 0$, $Y^b > 0$ is assumed, then the production possibility set can be defined as follows:

$$P = \left\{ \frac{x, y^g, y^b}{x} \geq X\lambda, y^g \geq Y^g\lambda, y^b \geq Y^b\lambda, \lambda > 0 \right\} \quad (8)$$

The SBM-Undesirable model is as follows:

$$\rho = min \frac{1 - \frac{1}{m}\sum_{i=1}^{m}\frac{S_i^-}{X_{io}}}{1 + \frac{1}{S_1+S_2}\left(\sum_{\gamma=1}^{s_1}\frac{S_\gamma^g}{y_{\gamma0}^g} + \sum_{\gamma=1}^{s_2}\frac{S_\gamma^b}{y_{\gamma0}^b}\right)}$$

$$S.t. \begin{cases} x_0 = X\lambda + S^- \\ y_0^g = Y^g\lambda - S^g \\ y_0^b = Y^b\lambda + S^b \\ S^- \geq 0, S^g \geq 0, S^b \geq 0, \lambda \geq 0 \end{cases} \quad (9)$$

Based on the SBM-DEA, the Malmquist index is defined as follows:

$$TFPch = TPch \times EFFch$$
$$= \left[\frac{d_i^t(x_{t+1}, y_{t+1})}{d_i^{t+1}(x_{t+1}, y_{t+1})} \times \frac{d_i^t(x_t, y_t)}{d_i^{t+1}(x_t, y_t)}\right]^{\frac{1}{2}} \times \frac{d_i^{t+1}(x^{t+1}, y^{t+1})}{d_i^{t+1}(x^t, y^t)} \quad (10)$$

where *TFPch* represents the Malmquist productivity index denoting GTFP, *TPch* the technical progress index, and *EFFch* the technical efficiency index. MATLAB software was used to calculate the TFP of environmental regulation in an output-oriented and variable way. Input and output indicators are shown in Table 1.

**3.2.2. Environmental regulation.** Administrative environmental regulation (*Gover*) is the use of administrative orders to restrict the standards and technologies used in the production and emissions of enterprises. The many administrative regulatory policies currently in use in China include the "three-simultaneous" system of pollutant emission standards, deadlines for treatment and shutdown and environmental impact assessment. As noted by Li and Ramanathan [45], the use of environmental administrative punishment cases is indicated.

Given China's imperfect market mechanism, the two market regulation tools (*Market*) of pollutant discharge subsidies and pollution permit transactions are not fully effective. Sewage charges, however, have been collected for a long time in China, and the system

**Table 1. Evaluation index system for GTFP.**

| Vector | Indicator | Measure | Unit |
|---|---|---|---|
| Input | Capital | Capital stock of 1978 as base period calculated by perpetual inventory method | Yuan |
| | Labor | Number of employees in different provinces | number |
| | Energy | Total energy consumption | ton of standard coal equivalent |
| Desirable Output | GDP | Total GDP | $10^8$ Yuan |
| Undesirable Output | Wastewater | Industrial wastewater emissions per unit GDP | ton/$10^4$ Yuan |
| | Solid | Industrial solid waste amount per unit GDP | ton/$10^4$ Yuan |
| | Exhaust | Sulfur dioxide emissions per unit GDP | ton/$10^4$ Yuan |

implementation procedures are well-established and effective. The total sewage fee income of a district is therefore used as a measure of market-based environmental regulations.

Information-based environmental regulation (*Inform*) is important because public participation in environmental regulation is not compulsory and relies on voluntary participation in environmental protection and supervision. However, the relevant laws and regulations are not complete, and the channels for the public to report environmental demands to the government remain relatively basic. Currently, they include only environmental complaints and environmental petitions. This study therefore uses the total number of petitions and the number of visitors to perform the entropy method TOPSIS to measure public participation in environmental regulation.

**3.2.3. Control variables.** The control variables in this study are as follows. (1) Industrial structure (*su*) is expressed as the proportion of the added value of the secondary industry to the total GDP. (2) Human capital is measured by level of education (*edu*) and is the most direct carrier of technology. The higher the level of education, the deeper the understanding of sustainable development. This study uses the number of years of education per capita as a measure of human capital. (3) R&D activity, the core of technological innovation, is measured by the ratio of R&D expenditure to GDP. (4) Foreign direct investment (*fdi*) is measured as the proportion of foreign direct investment to GDP. (5) The degree of openness (*open*) is measured as the ratio of total imports and exports to GDP. (6) The level of urbanization (*urban*) is also measured. (7) The public budget (*gov*) reflects the degree of government dominance over the economy, measured as the proportion of government fiscal expenditure to GDP. (8) The level of informatization (*email*) measures the level of development of information technology.

This study uses panel data for the entire country (excluding Tibet but covering the eastern, central, and western regions) from 2015 to 2018. The GTFP measure is based on the data from the Malmquist index. The relevant data on environmental regulations are derived from the China Environmental Yearbook [46] from 2005 to 2018, and the control variable data are derived from the China Statistical Yearbook [47]. Before carrying out the empirical tests, we performed descriptive statistical analysis for each variable. Table 2 gives the results for the natural logarithm of each variable.

**Table 2. Descriptive statistics of variables.**

| Variable | Obs. | Mean | Std. Dev. | Min. | Max. |
|---|---|---|---|---|---|
| *gtfp* | 420 | 0.7890 | 0.7101 | 0.1609 | 2.8147 |
| *gover* | 420 | 0.0089 | 0.0022 | 0.0028 | 0.0182 |
| *market* | 420 | 10.5822 | 1.4107 | 0.000 | 13.6911 |
| *inform* | 420 | 0.1421 | 0.124 | 0.000 | 0.7728 |
| *gdpc* | 420 | 2.359 | 1.5273 | 0.3355 | 8.5954 |
| *su* | 420 | 0.9435 | 0.4934 | 0.4945 | 4.1656 |
| *fes* | 420 | 10.8479 | 8.0532 | 1.2843 | 48.6444 |
| *edu* | 420 | 8.5873 | 0.9925 | 6.0405 | 12.3891 |
| *rdgdp* | 420 | 0.0134 | 0.0104 | 0.0017 | 0.0601 |
| *fdigdp* | 420 | 0.0255 | 0.0262 | 0.000 | 0.2074 |
| *open* | 420 | 0.3464 | 0.4425 | 0.0168 | 1.891 |
| *urban* | 420 | 0.5047 | 0.1453 | 0.1389 | 0.8961 |
| *govgdp* | 420 | 0.208 | 0.0946 | 0.0305 | 0.6269 |
| *post* | 420 | 6.1081 | 0.9265 | 3.0974 | 8.8382 |

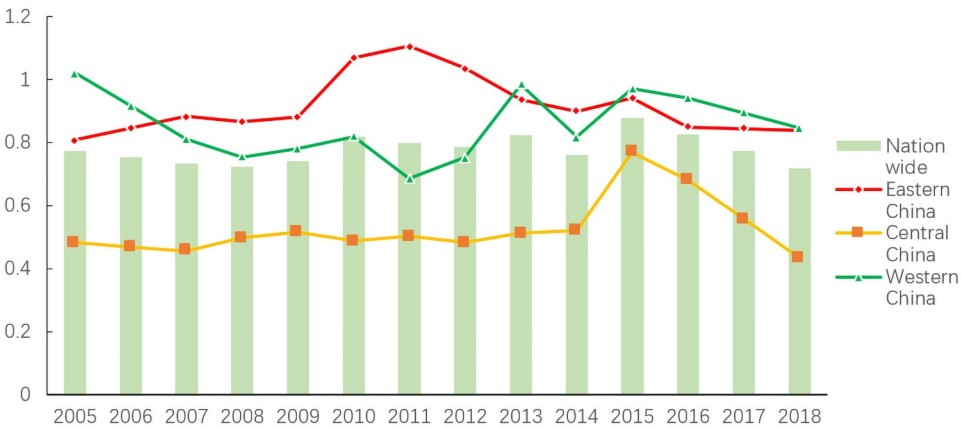

**Fig 1. Regional mean GTFP growth rate, 2005–2018.**

## 4. Analysis of empirical results

### 4.1 GTFP on the Malmquist productivity index

Fig 1 shows the trend of China's GTFP from 2005 to 2018. The average growth rate of GTFP dropped from 0.7743 in 2005 to 0.7169 in 2018, indicating an overall absence of growth during these 14 years. In terms of specific changes, GTFP decreased from 2005 to 2008, followed by a brief rise from 2009 to 2010 and further fluctuations until 2014, although these patterns are not very obvious. During the period of the 11th and 12th Five-Year Plans, the government adopted corporate energy-saving and emission reduction as a compulsory measure, and strengthened its responsibility for environmental pollution control. However, because of its long-term extensive economic development model and the 2008 financial shock, China has also strengthened the effects of investment and labor. As a result, inefficient production conditions caused by environmental pollution emissions have not been substantially improved. In 2015, the average value of China's GTFP rose to 0.8747, indicating that environmental supervision has achieved some positive results. However, since 2015, when China's economy began to show L-shaped economic growth, the pressure of growth caused a decline in GTFP, which fell to 0.7169 in 2018.

The eastern, central and western regions start from very different levels of economic development, and the changes to GTFP vary accordingly (Fig 2). Between 2007 and 2013, the eastern region had the highest level of GTFP and the fastest growth rate. Since 2013, the trend in the western region has been one of change. From 2005 to 2014, GTFP in the central region fluctuated little, showing steady development. In 2015, however, the growth rate of GTFP increased to 0.7695, after which it began to decline rapidly, reaching 0.4338 in 2018.

Since the reform and opening-up, the central region has developed its heavy industry in order to achieve economic growth. However, the region's human capital, management level and production technology remain limited, and the protection of environmental resources and application of penalties for environmental damage are relatively loose. This sacrifice of the environment in exchange for the rapid growth of economic aggregates is a low-level and unsustainable way to achieve economic growth. Despite the strengthening of environmental governance in 2015, the region has been unable to move to a development path less characterized by high pollution and high consumption.

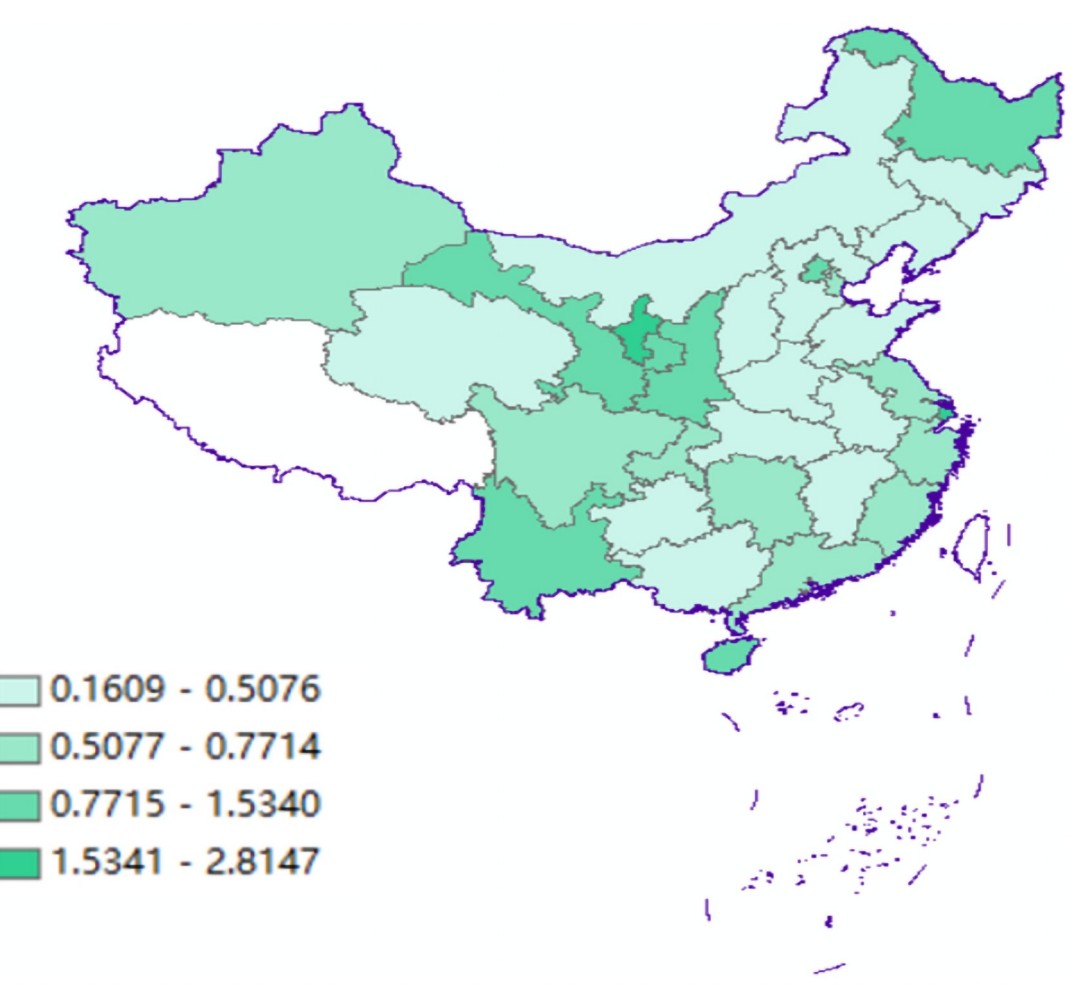

**Fig 2. Regional GTFP growth rate distribution, 2005–2018.**

The western region is rich in energy and has great potential for development. With the implementation of the Western Development Strategy and the emergence of latecomer advantages, the developed region has seen rapid growth in total economic volume. This will inevitably bring about greater consumption of natural resources and discharge of environmental pollutants, leading to poorer performance in GTFP.

## 4.2 Impact of environmental regulation on GTFP

This study uses STATA16.1 software for empirical analysis. It assumes that the proposed command, market and public participation types have a U-shaped relationship with GTFP, taking into account the resource endowments, industrial structure and development. There are differences between stages, and the impact of environmental regulations on GTFP may have different characteristics in the U-shaped structure. Therefore, it is necessary to investigate whether the proposed hypothesis is supported for the eastern, central and western regions on the basis of a national analysis.

There may be endogeneity problems in model estimation, so we use the system generalized method of moments (GMM) method. According to the estimation test results, the model has

no second-order sequence autocorrelation. The results of the Hansen test show that the instrumental variables estimated by GMM are valid. Table 3 shows the overall regression results for the impact of environmental regulations on GTFP.

For the whole country, whether the regulation is command-type, market-type or information-type, the impact of GTFP on the first-order coefficient is significantly positive and on the second-order coefficient significantly negative; that is, the impact of environmental regulation on GTFP is an inverted U-shaped structure. Before the inflection point, the increase in the intensity of administrative environmental regulation has a positive effect on the growth of GTFP, and the rate of increase slows gradually. After the inflection point, the increase in the intensity of administrative environmental regulation has a negative impact on GTFP, and the speed of reduction accelerates.

It is not difficult to find differences between the three inflection points of environmental regulation. In 2018, most regions crossed the threshold of government-type environmental regulation, which means that increasing government-type environmental regulation at this time had an adverse impact on GTFP. Most regions did not exceed the thresholds for market-based and information-based environmental regulations. For a long time, China's environmental regulations mainly took the form of government orders, but, with greater economic development, the market has played an increasingly important role in resource allocation. Too much emphasis on government regulations is not conducive to market resource allocation, and government-based environmental regulations have an adverse impact on GTFP. However, China's market and information-based regulatory systems are not complete, making it necessary to strengthen further the role of market-based and information-based environmental regulations.

From the perspective of control variables, in Model (1) the improvement in the industrial structure (*su*) is conducive to an improvement in GTFP. China's secondary industry already accounts for a high proportion of this and is still in a stage of rapid economic development. However, the continuous upgrading of the industrial structure in the central region has eliminated high-pollution and high-energy-consumption enterprises. The coefficient of human capital (*edu*) is estimated to be significantly positive, which means that improvements in human capital can increase the level of GTFP. On the one hand, human capital provides labor factor support for regional economic development; on the other hand, high-quality laborers have stronger environmental protection awareness and can play a role in social supervision. The research and development expenditure (*rd*) and foreign direct investment (*fdi*) coefficients are not significant. The level of openness (*open*) does not appear to affect the promotion of GTFP. At present, in the international division of labor, China is at the low end of the industrial chain. For a long time, the focus has been on labor-intensive and resource-intensive industries, and the opening-up of the region has disrupted the current industrial chain and exacerbated environmental pollution. The development of information technology (*email*) has not played a role in promoting the improvement of GTFP.

Given the differences in economic development between the eastern, central and western regions, we tested the relationship between different environmental regulations and the GTFP in those regions. The results are shown in Table 4.

The results in Table 4 show an inverted U-shaped relationship between different types of environmental regulations and GTFP in the eastern region. Information-based environmental regulation and government-commanded environmental regulation have the most significant impacts on the coefficient of GTFP, and the level of economic development in the eastern region is relatively high. It is the area where China's life insurance is subject to environmental regulations, and for a long time government-based environmental regulations have been the mainstay. With the continuous deepening of marketization and continuous improvements in

**Table 3. Regression results.**

| VARIABLES | (1) | (2) | (3) |
|---|---|---|---|
| | gtfp | gtfp | gtfp |
| L.gtfp | 0.4288*** | 0.4304*** | 0.4495*** |
| | (61.23) | (268.34) | (52.53) |
| govern | 1.1817*** | | |
| | (10.65) | | |
| govern2 | -0.6919*** | | |
| | (-10.60) | | |
| market | | 0.2900*** | |
| | | (2.66) | |
| market2 | | -0.0159*** | |
| | | (-2.99) | |
| inform | | | 1.2868*** |
| | | | (3.68) |
| inform2 | | | -2.0163*** |
| | | | (-2.58) |
| su | 0.0920*** | 0.0254 | -0.0625 |
| | (3.20) | (0.54) | (-0.71) |
| edu | 0.1023*** | 0.1156*** | 0.1229*** |
| | (9.03) | (14.54) | (11.58) |
| rdgdp | 1.3124 | -1.0218 | -2.1890 |
| | (0.39) | (-0.55) | (-1.10) |
| fdigdp | -0.5725 | 0.0664 | 3.1608 |
| | (-0.87) | (0.06) | (1.15) |
| open | -0.0811** | -0.1128*** | -0.1386** |
| | (-1.99) | (-3.04) | (-2.16) |
| urban | -0.0826 | 0.1459** | -0.0646 |
| | (-1.49) | (2.09) | (-0.85) |
| govgdp | -0.5563*** | -0.3069 | 0.2174 |
| | (-2.76) | (-1.58) | (0.44) |
| lab | -0.4114*** | -0.3612*** | -0.4290*** |
| | (-33.39) | (-25.04) | (-26.52) |
| Constant | 2.5684*** | 1.1837** | 2.8281*** |
| | (29.33) | (1.99) | (10.10) |
| ID/Year | Control | Control | Control |
| Observations | 390 | 390 | 390 |
| Number of IDs | 30 | 30 | 30 |
| Hansen | 23.40 | 23.55 | 23.63 |
| P-Hansen | [1.000] | [1.000] | [1.000] |
| AR(2) | 0.888 | 0.895 | 0.847 |
| P-AR(2) | [0.375] | [0.371] | [0.397] |
| Inflection point | 0.2928 | | |

Note: Significance levels of 1%, 5% and 10% are denoted by ***, ** and *, respectively. T values are given in parentheses and P values in square brackets.

**Table 4. Impact of environmental regulation on GTFP regression by region.**

| Variable | GTFP, eastern region | | | GTFP, central region | | | GTFP, western region | | |
|---|---|---|---|---|---|---|---|---|---|
| | (1) | (2) | (3) | (4) | (5) | (6) | (7) | (8) | (9) |
| L.gtfp | 0.2694*** | 0.2487*** | 0.4405*** | 0.7134*** | 0.0963 | 0.1890*** | 0.2243*** | 0.3192* | 0.6529*** |
| | (47.11) | (36.86) | (56.84) | (18.55) | (1.08) | (6.03) | (22.85) | (1.65) | (16.64) |
| govern | 0.9281*** | | | 1.4873*** | | | 0.9763* | | |
| | (3.59) | | | (3.12) | | | (1.89) | | |
| govern2 | -0.6528*** | | | -0.7526*** | | | -0.4990* | | |
| | (-4.29) | | | (-3.00) | | | (-1.80) | | |
| market | | 0.1260* | | | 0.9282** | | | -0.7450* | |
| | | (1.89) | | | (2.05) | | | (-1.79) | |
| market2 | | -0.0076** | | | -0.0519** | | | 0.0346* | |
| | | (-2.15) | | | (-2.31) | | | (1.78) | |
| inform | | | 0.8616* | | | -1.0602** | | | 2.5685*** |
| | | | (1.83) | | | (-2.19) | | | (4.80) |
| inform2 | | | -0.9839* | | | 2.3157* | | | -7.5011*** |
| | | | (-1.70) | | | (1.70) | | | (-4.59) |
| ID/Year | Control | Control | Control | Control | Control | Control | Control | Control | Control |
| Observations | 152 | 152 | 152 | 118 | 120 | 120 | 120 | 118 | 118 |
| Number of IDs | 30 | 30 | 30 | 22 | 28 | 28 | 28 | 22 | 22 |
| Hansen | 12.72 | 17.11 | 18.15 | 13.26 | 13.60 | 16.22 | 16.22 | 6.41 | 11.11 |
| P-Hansen | [1.000] | [1.000] | [1.000] | [1.000] | [1.000] | [1.000] | [1.000] | [1.000] | [1.000] |
| AR(2) | 0.86 | 0.85 | 0.92 | 0.97 | 0.67 | 0.68 | 0.70 | -0.38 | 1.42 |
| P-AR(2) | [0.391] | [0.395] | [0.358] | [0.333] | [0.504] | [0.498] | [0.484] | [0.702] | [0.155] |

Note: Significance levels of 1%, 5% and 10% are denoted by ***, ** and *, respectively. T values are given in parentheses and P values in square brackets.

the quality of the labor force in the east, social supervision has begun to play a role in improving GTFP.

The government's environmental regulations and market-based environmental regulations have a U-shaped relationship with GTFP, but information-based environmental regulations and GTFP have an inverted U-shaped relationship. For the central region, environmental regulation takes the form of executive orders from the central government, and unswerving support is therefore required. Although the economic development level of the central region lags behind that of the eastern region, with the acceptance of the industrial transfer of the eastern region and its huge population and resource dividends, the environmental regulation of the central region draws effectively on high levels of experience. Because human capital lags behind in the eastern region, huge resource dividends and residents' awareness of the importance of environmental protection have led to a lack of social supervision.

Command-type and information-type environmental regulations in the western region have an inverted U-shaped relationship with GTFP, but market-type environmental regulations have a U-shaped relationship. The reason for the inverted U-shape of government-ordered regulation is similar to that in central China: implementation of environmental regulatory orders from central government. However, the western region lags behind the central region in its levels of economic development and human capital. The market-oriented environmental regulations cannot effectively promote an improvement in GTFP, and marketability remains very low. In contrast, with the implementation of the Western Development

Strategy and the development of a western ecological environment protection strategy, the Chinese government has cultivated residents' environmental awareness, avoided methods of economic development that sacrifice the environment, and ensured that information-based supervision is effective.

## 5. Conclusions and policy recommendations

In order to examine the impact of environmental regulation on GTFP, this study first divided environmental regulation into three types (administrative, market-based and information-based) and proposed relevant research assumptions based on different types of environmental regulation tools. Second, it used the SBM-DEA model to measure GTFP and its composition, both in China as a whole and in the eastern, central and western regions, from 2005 to 2018. It analyzed and compared the changes and growth drivers of GTFP in the different regions. Third, using panel regression models and starting from the overall level and the three regional levels, it conducted an empirical analysis of the relationships between the three types of environmental regulations and GTFP. The main research conclusions are as follows.

First, GTFP has changed significantly, both across the country and in the eastern, central and western regions. In terms of growth of GTFP, the east region ranks highest of the three, and the central region lowest. The main reason for the growth of GTFP in the eastern region is its relatively high level of economic development, which leads to relatively high levels of environmental governance technology and environmental governance investment. The western region enjoys transfer payments for environmental protection from central government, and therefore its degree of ecological environmental protection is better. The economic development of the central region lags behind that of the eastern region, and its ecological environment lags behind that of the western region. Huge population and economic development pressures are the main reasons for the slow growth of GTFP in the central region.

Second, there is a non-linear relationship between environmental regulations and GTFP. The impact of imperative environmental regulation on GTFP for China overall and for the regions is generally U-shaped. The intensity of environmental regulation in most regions has not crossed the inflection point, and it still appears to promote GTFP. The role of market-based regulatory tools on GTFP has an inverted U-shaped structure in China overall and in the eastern and central regions, but a U-shaped relationship in the western region. The impact of information-based environmental regulation on GTFP has a positive U-shaped structure in the western region but has not crossed the inflection point for China as a whole, and it has a U-shaped relationship in the central region.

In order to promote the continuous growth of GTFP with the help of environmental regulations, this study makes the following policy recommendations on the basis of its empirical results. First, when selecting environmental regulatory tools, government should consider the level of regional economic development and how the tools will be used. This is necessary to benefit fully from the advantages of various environmental regulation tools and to limit the intensity of environmental regulations to a reasonable level. Second, it is important to pay attention to innovation and optimization of environmental regulatory tools, particularly so that administrative tools can be made compatible with other types. Third, the environmental regulatory policy system should be revised and improved in a timely manner. The openness and timeliness of environmental information should be enhanced, and a standardized environmental regulatory social supervision system should be implemented. Finally, green technology innovation should be encouraged by increasing subsidies for technology research and development, and by reducing taxes and fees on green products.

## Author Contributions

**Conceptualization:** Qin He.

**Data curation:** Yaowu Han.

**Formal analysis:** Qin He.

**Funding acquisition:** Qin He.

**Investigation:** Qin He, Lei Wang.

**Methodology:** Qin He, Yaowu Han.

**Project administration:** Qin He.

**Resources:** Yaowu Han.

**Software:** Yaowu Han.

**Supervision:** Qin He.

**Validation:** Lei Wang.

**Visualization:** Qin He.

**Writing – original draft:** Qin He.

**Writing – review & editing:** Qin He, Lei Wang.

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
