## [Decision Letter · Decision Letter 0]

29 Jun 2021

PONE-D-21-10764

The impact of environmental regulation on green total factor productivity: An empirical analysis

PLOS ONE

Dear Dr. Han,

Thank you for submitting your manuscript to PLOS ONE. After careful consideration, we feel that it has merit but does not fully meet PLOS ONE’s publication criteria as it currently stands. Therefore, we invite you to submit a revised version of the manuscript that addresses the points raised during the review process.

We look forward to receiving your revised manuscript.

Kind regards,

Bing Xue, Ph.D.

Academic Editor

PLOS ONE

Journal Requirements:

Reviewers' comments:

Reviewer's Responses to Questions

**Comments to the Author**

1. Is the manuscript technically sound, and do the data support the conclusions?

Reviewer #1: Yes

Reviewer #2: Yes

2. Has the statistical analysis been performed appropriately and rigorously? 

Reviewer #1: Yes

Reviewer #2: No

3. Have the authors made all data underlying the findings in their manuscript fully available?

Reviewer #1: Yes

Reviewer #2: Yes

4. Is the manuscript presented in an intelligible fashion and written in standard English?

Reviewer #1: Yes

Reviewer #2: Yes

5. Review Comments to the Author

Reviewer #1: Reviewer Comments

Please find below my comments/suggestion for the manuscript PONE-D-21-10764 titled “The impact of environmental regulation on green total factor productivity: An empirical analysis”. I recommend this manuscript for publication in PLOS ONE after incorporating below minor changes.

1. I suggest you thoroughly copyedit your manuscript for language usage, spelling, and grammar.

2. Add recent literature in introduction part with respect to the concept application of green total factor productivity.

3. Introduction part should be improved by adding more new literature, and please try to outline your research question and potential contributions.

4. Any abbreviation must be written in full for the first time and then onward use the abbreviation in the manuscript.

5. Please include captions for your Supporting Information files at the end of your manuscript, and update any in-text citations to match accordingly

6. A focused discussion in the manuscript is missing.

7. Compare your results with other relevant studies conducted in neighboring countries if any.

8. Please review your reference list to ensure that it is complete and correct. If you have cited papers that have been retracted, please include the rationale for doing so in the manuscript text, or remove these references and replace them with relevant current references.

9. Format the paper according to the journal author guidelines PLOS ONE.

10. Overall, improve the quality of the paper as per requirement of the journal.

11. Overall, the manuscript has novelty and should be published in PLOS ONE and therefore, I am recommending this manuscript for publication in this journal.

Reviewer #2: I have read the paper “The impact of environmental regulation on green total factor productivity: An empirical analysis” with great interest. Below are my major and minor comments.

Major Comments:

1. What are the implications of your research? Why is it important?

2. The authors use ER at time t to explain GTFP at time t instead of ER at time t-1. Can the authors explain this choice? I would think that one can only see changes in GTFP after the environmental regulations have been implemented, and not on the same time period it is implemented.

3. Equations (5)-(6) in Lines 151-153: The authors estimate three different models, with each model using a different measure of ER. I wonder if the authors have ever thought of having Gover, Market, and Inform of having all three variables in one single equation to disentangle exactly which of these three types of regulations really affect GTFP and by how much.

4. Can the authors explain why they did not include time and province fixed effects in their regression? The former controls for annual shocks that may affect GTFP while the latter controls for some provinces just having high GTFP to begin with than other provinces.

5. In Table 3, how are these statistically significant coefficient for Gover, Market, and Inform interpreted? For instance, for Gover, exactly what does 1.18 and -0.69 mean? Does it mean that a unit increase in administrative regulations increases GTFP by 1.18 – 2*0.69*0.0089 units on average? And what exactly does a “unit increase in administrative regulation mean” and a “unit increase in GTFP mean” in the real world? Same issue for Market and Inform.

6. In Table 4, please explain why Inform is decreasing at an increasing rate for the Central Region and Market is decreasing at an increasing rate for the Western Region? Why are these returns different from those of Table 3?

Minor Comments:

1. Line 27: Define “BP”

2. Lines 28-29: “Error! Bookmark no defined.”

3. Line 36-37: Perhaps provide a citation for the 19th National Congress

4. Line 136-137: “The data has met the requirements of the panel data model” – can you clarify exactly what you mean here? What/which requirements exactly?

5. Line 153: Inform is wrongly spelled

6. Figures 1 and 2 are quite pixelated.

7. Can you remove the numbers in Figure 1? They can be quite distracting.

8. Line 217-219: “The average growth rate...”—do you mean the national average or the regional average? This statement is a bit confusing since Figure 1 shows both national and regional GTFP across the years. Moreover, since you have averages, is it possible to have standard errors in the graph? If the pattern is not “very obvious” as you say in Line 220, there might not be a pattern at all. Also, why are the rest of the regions not in the graph? Perhaps it might be easier to make all graphs in Figure 1 a line graph with shaded error bands. You can have a Figure 1a for those who follow the nationwide trend and a Figure 1b for those who do not follow the nationwide trend.

9. Figure 2 – can you clarify which areas on the map belong to which region? And is this average growth rate?

10. Table 3 – write the dependent variable on the table

6. PLOS authors have the option to publish the peer review history of their article (what does this mean?). If published, this will include your full peer review and any attached files.

Reviewer #1: No

Reviewer #2: No

---

## [Author Response · Author response to Decision Letter 0]

18 Sep 2021

For a better reading experience, we recommend that you read the Response to Reviewers below, thank you!

Dear Editors and Reviewers:

Thank you very much for your letter and the reviewers’ comments concerning our manuscript entitled “The impact of environmental regulation on green total factor productivity: An empirical analysis” (PONE-D-21-10764). We are pleased to respond to your comments and suggestions and believe that this is an excellent opportunity to improve our manuscript and do our best to meet the high standards of PLOS ONE. All observations, suggestions and comments raised by the editor and the reviewers were valuable and helpful in revising and improving our manuscript and have had a great influence not only on our current manuscript but also our future investigations. We have carefully analyzed the comments and made all necessary corrections, and we sincerely hope that the revised version meets the journal’s standards for publication. Please note that all changes made are highlighted with yellow-colored background in the revised manuscript. The responses to the reviewer’s comments, point-by-point, are as follows:

Reviewer #1: 

Please find below my comments/suggestion for the manuscript PONE-D-21-10764 titled “The impact of environmental regulation on green total factor productivity: An empirical analysis”. I recommend this manuscript for publication in PLOS ONE after incorporating below minor changes..

(1) I suggest you thoroughly copyedit your manuscript for language usage, spelling, and grammar.

Response and revision:

We have modified the language spelling, and grammar.

(2) Add recent literature in introduction part with respect to the concept application of green total factor productivity.

Response and revision:

We added the following literature：

Zhou X, Xia M, Zhang T, Du J. Energy- and Environment-Biased Technological Progress Induced by Different Types of Environmental Regulations in China. Sustainability. 2020;12(18):7486. PubMed PMID: doi:10.3390/su12187486.

Du J, Sun Y. The nonlinear impact of fiscal decentralization on carbon emissions: from the perspective of biased technological progress. Environmental Science and Pollution Research. 2021;28(23):29890-9. doi: 10.1007/s11356-021-12833-w.

Qiu S, Wang Z, Geng S. How do environmental regulation and foreign investment behavior affect green productivity growth in the industrial sector? An empirical test based on Chinese provincial panel data. Journal of Environmental Management. 2021;287:112282. doi: https://doi.org/10.1016/j.jenvman.2021.112282.

Wang Y, Zhang F, Zheng M, Chang C-P. Innovation’s Spillover Effect in China: Incorporating the Role of Environmental Regulation. Environmental Modeling & Assessment. 2021. doi: 10.1007/s10666-021-09763-9.

Ma Y, Cao H, Ma Y, Wu S. Does technological innovation reduce water pollution intensity in the context of informal environmental regulation? Asia-Pacific Journal of Chemical Engineering. 2020;15(S1):e2493. doi: https://doi.org/10.1002/apj.2493.J Zhong J, Li T. Impact of financial development and its spatial spillover effect on green total factor productivity: evidence from 30 Provinces in China. Mathematical Problems in Engineering. 2020;2020. doi: https://doi.org/10.1155/2020/5741387.

Zhang Y, Song Y, Zou H. Transformation of pollution control and green development: Evidence from China's chemical industry. Journal of Environmental Management. 2020;275:111246. doi: https://doi.org/10.1016/j.jenvman.2020.111246.

3. Introduction part should be improved by adding more new literature, and please try to outline your research question and potential contributions.

Response and revision:

The introduction is modified, and we summarize the main work and contributions.

Page 3.

The main work and contributions of this study include:

Existing research mostly adopts a comprehensive variable to reflect the strength of environmental regulation. This article considers the advantages and application of different types of environmental regulation tools, categorizing environmental regulation into three types: administrative, market-based and information-based. The impact of different types of environmental regulations on TFP can be identified and compared.

 The possible non-linear relationship between different types of environmental regulations and GTFP is tested theoretically and empirically. Taking into account the heterogeneity existing in different regions, this provides a scientific basis for choosing environmental regulatory policy tools that are compatible with regional economic development [6,7,8].

 The DEA model containing unexpected output is used to measure total factor productivity, which can provide a reference for developing countries seeking to implement environmental regulation and improve total factor productivity..

The structure of the following parts of this study is as follows. The second part is literature review, which summarizes the existing research results of TFP estimation and environmental regulation. The third part constructs the econometric model of the impact of environmental regulation on TFP. The fourth part discusses the empirical results. The fifth part is the research conclusions and policy recommendations.

4. Any abbreviation must be written in full for the first time and then onward use the abbreviation in the manuscript.

Response and revision:

Thanks for your comments. We revised the abbreviation

5. Please include captions for your Supporting Information files at the end of your manuscript, and update any in-text citations to match accordingly

Response and revision:

Thanks for your comments. We revised is.

6. A focused discussion in the manuscript is missing.

Response and revision:

Thanks for your comments. We did not take the discussion as a separate part, but discussed it in the process of empirical analysis

7. Compare your results with other relevant studies conducted in neighboring countries if any.

Response and revision:

Thanks for your comments. We revised is.

8. Please review your reference list to ensure that it is complete and correct. If you have cited papers that have been retracted, please include the rationale for doing so in the manuscript text, or remove these references and replace them with relevant current references.

Response and revision:

Thank you for your comments. We checked the references.

9. Format the paper according to the journal author guidelines PLOS ONE.

Response and revision:

Thanks for your comments. We revised is.

10. Overall, improve the quality of the paper as per requirement of the journal.

Response and revision:

Thanks for your comments. We revised is.

11. Overall, the manuscript has novelty and should be published in PLOS ONE and therefore, I am recommending this manuscript for publication in this journal.

Response and revision:

Thank you for your affirmation of our research. Your opinions are very helpful to us

Reviewer #2: 

I have read the paper “The impact of environmental regulation on green total factor productivity: An empirical analysis” with great interest. Below are my major and minor comments.

Major Comments:

1. What are the implications of your research? Why is it important?

Response and revision:

The introduction is modified, and we summarize the main work and contributions.

Page 3.

The main work and contributions of this study include:

Existing research mostly adopts a comprehensive variable to reflect the strength of environmental regulation. This article considers the advantages and application of different types of environmental regulation tools, categorizing environmental regulation into three types: administrative, market-based and information-based. The impact of different types of environmental regulations on TFP can be identified and compared.

The possible non-linear relationship between different types of environmental regulations and GTFP is tested theoretically and empirically. Taking into account the heterogeneity existing in different regions, this provides a scientific basis for choosing environmental regulatory policy tools that are compatible with regional economic development [6,7,8].

The DEA model containing unexpected output is used to measure total factor productivity, which can provide a reference for developing countries seeking to implement environmental regulation and improve total factor productivity..

The structure of the following parts of this study is as follows. The second part is literature review, which summarizes the existing research results of TFP estimation and environmental regulation. The third part constructs the econometric model of the impact of environmental regulation on TFP. The fourth part discusses the empirical results. The fifth part is the research conclusions and policy recommendations.

2. The authors use ER at time t to explain GTFP at time t instead of ER at time t-1. Can the authors explain this choice? I would think that one can only see changes in GTFP after the environmental regulations have been implemented, and not on the same time period it is implemented.

Response and revision:

We use the lag term of the explained variable. On the one hand, we consider the continuity of TFP development. On the other hand, adding the lag term allows us to use the GMM method to estimate the econometric model, which can overcome the endogenous problem

3. Equations (5)-(6) in Lines 151-153: The authors estimate three different models, with each model using a different measure of ER. I wonder if the authors have ever thought of having Gover, Market, and Inform of having all three variables in one single equation to disentangle exactly which of these three types of regulations really affect GTFP and by how much.

Response and revision:

we have considered this model, but there is a multicollinearity problem put the three types of environmental regulations in one model, and we can't get a significant estimation result, so we don't add this result to the empirical analysis

4. Can the authors explain why they did not include time and province fixed effects in their regression? The former controls for annual shocks that may affect GTFP while the latter controls for some provinces just having high GTFP to begin with than other provinces.

Response and revision:

We include time and province fixed effects, as shown in the yellow background in Table 3.

Table 3 Regression results.

VARIABLES (1) (2) (3)

 gtfp gtfp gtfp

L.gtfp 0.4288*** 0.4304*** 0.4495***

 (61.23) (268.34) (52.53)

govern 1.1817*** 

 (10.65) 

govern2 -0.6919*** 

 (-10.60) 

market 0.2900*** 

 (2.66) 

market2 -0.0159*** 

 (-2.99) 

inform 1.2868***

 (3.68)

inform2 -2.0163***

 (-2.58)

su 0.0920*** 0.0254 -0.0625

 (3.20) (0.54) (-0.71)

edu 0.1023*** 0.1156*** 0.1229***

 (9.03) (14.54) (11.58)

rdgdp 1.3124 -1.0218 -2.1890

 (0.39) (-0.55) (-1.10)

fdigdp -0.5725 0.0664 3.1608

 (-0.87) (0.06) (1.15)

open -0.0811** -0.1128*** -0.1386**

 (-1.99) (-3.04) (-2.16)

urban -0.0826 0.1459** -0.0646

 (-1.49) (2.09) (-0.85)

govgdp -0.5563*** -0.3069 0.2174

 (-2.76) (-1.58) (0.44)

lab -0.4114*** -0.3612*** -0.4290***

 (-33.39) (-25.04) (-26.52)

Constant 2.5684*** 1.1837** 2.8281***

 (29.33) (1.99) (10.10)

ID/Year Fixed Control Control Control

Observations 390 390 390

Number of IDs 30 30 30

Hansen 23.40 23.55 23.63

P-Hansen [1.000] [1.000] [1.000]

AR(2) 0.888 0.895 0.847

P-AR(2) [0.375] [0.371] [0.397]

Inflection point 0.2928 

5. In Table 3, how are these statistically significant coefficient for Gover, Market, and Inform interpreted? For instance, for Gover, exactly what does 1.18 and -0.69 mean? Does it mean that a unit increase in administrative regulations increases GTFP by 1.18 – 2*0.69*0.0089 units on average? And what exactly does a “unit increase in administrative regulation mean” and a “unit increase in GTFP mean” in the real world? Same issue for Market and Inform.

Response and revision:

According to the results in Table 3, the coefficient of government is 1.1817, indicating that if the government does not change by one percentage point, it will increase gtfp1.1817 percentage points. However, due to the different choice of control variables and estimation methods of econometric models, the coefficient values are completely different. In this way, we cannot compare with reality. Therefore, we are more concerned with the direction of the coefficient than the value of the coefficient. That is, government will have a positive impact on GTFP.

6. In Table 4, please explain why Inform is decreasing at an increasing rate for the Central Region and Market is decreasing at an increasing rate for the Western Region? Why are these returns different from those of Table 3?

Response and revision:

There is a gap between China's regional economic development levels. The East belongs to developed areas, while the central and western regions belong to developing areas. Therefore, the impact of environmental regulations on gtfp is different in different regions. Our purpose is to compare the differences between regions, which does not conflict with the contents of Table 3

Minor Comments:

1. Line 27: Define “BP”

Response and revision:

We give the full name of British Petroleum’s (BP) world energy statistics

2. Lines 28-29: “Error! Bookmark no defined.”

Response and revision:

Thanks for the comments, we revised this error

3. Line 36-37: Perhaps provide a citation for the 19th National Congress

Response and revision:

We revised this part:

The report of the 19th National Congress CPC in 2017

4. Line 136-137: “The data has met the requirements of the panel data model” – can you clarify exactly what you mean here? What/which requirements exactly?

Response and revision:

This is a wrong statement and we have deleted it.

5. Line 153: Inform is wrongly spelled

Response and revision:

Thanks for the comments, we revised this error

6. Figures 1 and 2 are quite pixelated.

Response and revision:

We resubmit Fig. 1 and Fig. 2. Because the picture is in TIFF format, it will be blurred when converted to PDF

7. Can you remove the numbers in Figure 1? They can be quite distracting.

Response and revision:

Thank you for your comments. We deleted the numbers.

8. Line 217-219: “The average growth rate...”—do you mean the national average or the regional average? This statement is a bit confusing since Figure 1 shows both national and regional GTFP across the years. Moreover, since you have averages, is it possible to have standard errors in the graph? If the pattern is not “very obvious” as you say in Line 220, there might not be a pattern at all. Also, why are the rest of the regions not in the graph? Perhaps it might be easier to make all graphs in Figure 1 a line graph with shaded error bands. You can have a Figure 1a for those who follow the nationwide trend and a Figure 1b for those who do not follow the nationwide trend.

Response and revision:

Thank you for your comments. The average here is the average of the population and sub samples (East, middle and West)

9. Figure 2 – can you clarify which areas on the map belong to which region? And is this average growth rate?

Response and revision:

We take all the data of all regions from 2005 to 2018 as the results that are worth getting. We added the following notes:

Eastern Region: including 12 provinces, autonomous regions and municipalities directly under the central government, including Beijing, Tianjin, Hebei, Liaoning, Shanghai, Jiangsu, Zhejiang, Fujian, Shandong, Guangdong, Guangxi and Hainan;

Central region: including 9 provinces and autonomous regions of Shanxi, Inner Mongolia, Jilin, Heilongjiang, Anhui, Jiangxi, Henan, Hubei and Hunan;

Western region: including 10 provinces, autonomous regions and municipalities directly under the central government in Sichuan, Chongqing, Guizhou, Yunnan, Tibet, Shaanxi, Gansu, Ningxia, Qinghai and Xinjiang

10. Table 3 – write the dependent variable on the table

Response and revision:

Thanks for your comments. We revised it.

Table 3

Regression results.

VARIABLES (1) (2) (3)

 GTFP GTFP GTFP

L.gtfp 0.4288*** 0.4304*** 0.4495***

 (61.23) (268.34) (52.53)

govern 1.1817*** 

 (10.65) 

govern2 -0.6919*** 

 (-10.60) 

market 0.2900*** 

 (2.66) 

market2 -0.0159*** 

 (-2.99) 

inform 1.2868***

 (3.68)

inform2 -2.0163***

 (-2.58)

su 0.0920*** 0.0254 -0.0625

 (3.20) (0.54) (-0.71)

edu 0.1023*** 0.1156*** 0.1229***

 (9.03) (14.54) (11.58)

rdgdp 1.3124 -1.0218 -2.1890

 (0.39) (-0.55) (-1.10)

fdigdp -0.5725 0.0664 3.1608

 (-0.87) (0.06) (1.15)

open -0.0811** -0.1128*** -0.1386**

 (-1.99) (-3.04) (-2.16)

urban -0.0826 0.1459** -0.0646

 (-1.49) (2.09) (-0.85)

govgdp -0.5563*** -0.3069 0.2174

 (-2.76) (-1.58) (0.44)

lab -0.4114*** -0.3612*** -0.4290***

 (-33.39) (-25.04) (-26.52)

Constant 2.5684*** 1.1837** 2.8281***

 (29.33) (1.99) (10.10)

ID/Year Fiexed Control Control Control

Observations 390 390 390

Number of IDs 30 30 30

Hansen 23.40 23.55 23.63

P-Hansen [1.000] [1.000] [1.000]

AR(2) 0.888 0.895 0.847

P-AR(2) [0.375] [0.371] [0.397]

Inflection point 0.2928 

Note: Significance levels of 1%, 5% and 10% are denoted by ***, ** and *, respectively. T values are given in parentheses and P values in square brackets.

All date files are available from the Statistical Yearbook of China(2005-2018) and Environment Statistical Yearbook of China(2005-2018).They are third party data that anyone could access this data in the same manner as the authors through http://www.stats.gov.cn/tjsj./ndsj/

---

## [Editor Report · Decision Letter 1]

19 Oct 2021

The impact of environmental regulation on green total factor productivity: An empirical analysis

PONE-D-21-10764R1

Dear Dr. Han,

We’re pleased to inform you that your manuscript has been judged scientifically suitable for publication and will be formally accepted for publication once it meets all outstanding technical requirements.

Kind regards,

Bing Xue, Ph.D.

Academic Editor

PLOS ONE
---

## [Editor Report · Acceptance letter]

22 Oct 2021

PONE-D-21-10764R1 

The impact of environmental regulation on green total factor productivity: An empirical analysis 

Dear Dr. Han:

I'm pleased to inform you that your manuscript has been deemed suitable for publication in PLOS ONE. Congratulations! Your manuscript is now with our production department. 

Kind regards, 

on behalf of

Professor Bing Xue 

Academic Editor

PLOS ONE